# Consumer Evaluation of Agricultural Products Produced in Areas Affected by Natural Disasters: A Case Study of Damaged Apples in Japan

**DOI:** 10.3390/foods12132498

**Published:** 2023-06-27

**Authors:** Eriko Miyama, Tamaki Morita

**Affiliations:** 1Faculty of Agriculture, Tokyo University of Agriculture and Technology, 3-5-8 Saiwai-cho, Tokyo 183-8509, Japan; 2Faculty of Aviation Management, J. F. Oberlin University, 2-31-1 Ochiai, Tokyo 206-0033, Japan

**Keywords:** consumer evaluation, natural disaster, conjoint analysis, damaged apples, sustainable agriculture, Japanese co-ops

## Abstract

This study examines how consumers perceive agricultural products affected by natural disasters, using apples in Japan as a case study. Typhoons and other natural disasters frequently damage the surface of apples during the production season, causing significant harm to farmers’ businesses, particularly when a large typhoon hits the production area. To maintain the sustainability of agricultural production, consumers need to purchase damaged crops at a certain price. To assess the effect of product attributes, such as appearance and price, and personal attributes of respondents, we conducted a choice-based conjoint analysis using a mixed logit model. The estimated results using the main effect and cross-section models show that consumers generally devalue apples when they have scratches. However, by using consumer co-operatives on a daily basis and disseminating simple information about the relationship between scratches and natural disasters, we could mitigate this devaluation tendency and thus contribute to sustainable agricultural production.

## 1. Introduction

In recent years, the global supply and demand of agricultural products has been disrupted by natural disasters, such as earthquakes, tsunamis, and typhoons. These disasters often result in damaged products, which are typically (1) sold to distributors or consumers at a low price, (2) discarded, or (3) sold to processors as raw materials. However, when a large number of products are damaged due to a disaster, it is important for producers to have the original product purchased at a reproducible price, as the unit price for raw materials is set at a lower level than that of the original product. Consumer understanding of the instability of agricultural production is crucial for sustainable agricultural reproduction, particularly in light of production instability and natural disasters.

Unfortunately, consumers are often uninformed about agricultural production processes, including the impacts of disaster, and tend to rely on appearance as the most readily available information for evaluation when purchasing fresh products. Supportive and ethical consumption practices, which actively inform consumers about disasters and encourage them to make purchases, have recently gained attention. Ethical consumerism, as defined by Kushwah et al. [1] (p. 1), refers to a form of “consumer activism focusing on the production and consumption of products based on social and environmental concerns”. To encourage ethical consumption, it is essential to provide consumers with information about the production of goods. However, significant information asymmetry between consumers and producers in the distribution of agricultural products makes it challenging for consumers to obtain accurate and detailed information about production.

One way to bridge the information gap between producers and consumers is through the use of consumer co-operatives (co-ops) that provide consumer education. In Japan, co-ops that provide direct-to-consumer services are popular. By using co-ops, consumers can access information related to production on a daily basis. For example, co-ops are making efforts to eliminate information asymmetry between producers and consumers by visiting production sites, disclosing production histories, relaying messages from producers, and collecting and publicizing consumers’ opinions. This is because the organizations that have promoted co-op or produce-direct alliances in Japan are positioned as one of the alternative food networks against the large food industry networks, focusing on the morality of consumption, changing food habits, and the formation of consumer communities that promote alternative food supplies [2,3].

The first consumer co-operatives were established in Japan around the end of the 19th century. In the late 1940s, after World War II, consumer co-operatives were spread across the country as a consumer-led collective movement in response to the consumers’ needs. Later, during the period of rapid economic growth in the 1970s, consumer co-operatives promoted the involvement of consumers in joint-buying and supermarket businesses as a method of ensuring food safety and protecting consumers from rapid inflation. During this time, co-ops expanded their membership through the development of safe food products and direct-to-consumer services [4] (p. 11).

The main activity of co-ops is the sale of food and household goods, but they also provide a wide range of other consumer services, such as insurance and nursing care. In order to use the delivery and joint-buying services, consumers must become members of the co-operative. Members are also entitled to information on other services, such as nursing care, insurance, housing and learning on agricultural productions. Typically, a delivery fee or commission is charged for each delivery, but there is no membership fee. Instead, members pay a specified amount of money in the form of a capital contribution at the time of joining or during the term of membership, which is refunded when they leave the co-operative; dividends are then distributed to members according to the amount of their contributions if the co-operative earns a high profit.

Co-ops are actively engaged in sales activities, regularly distributing sample products to local households to recruit members, and as of 2022, about 39% of all households in Japan held co-op subscriptions [4] (p. 15). Since it is possible to purchase items in-person from the stores without becoming a member, the percentage of the population actually using co-ops is likely to be higher than this. Many co-ops also claim to contribute to food safety and sustainable agricultural production, and provide opportunities for consumers to learn about agricultural production through visits to production areas, study meetings, and exchanges of opinions [4] (p. 9).

Despite efforts to promote ethical consumption, the issue of high prices for ethical products remains a concern in actual sales settings, leading some consumers to avoid purchasing such products [5,6,7,8]. Yomoah [8] pointed that in addition to price, everyday consumption trends are an obstacle to ethical consumption. This indicates that daily use of co-ops might mitigate this tendency. Additionally, many consumers place a higher value on visual appearance and may therefore view blemished products as less desirable, potentially affecting their willingness to pay (WTP) for them [9,10].

Most of the research on ethical consumption focuses on the value added to production or distribution processes; the working conditions of producers, such as fair trade, are considered, as is production that is environmentally friendly, such as organic production [1,11]. Disaster-affected agricultural products have not been the main focus of these ethical consumption analyses. However, supportive consumption is receiving attention with the recent increase in natural disasters in production areas, especially food ingredient production centers, following the Great East Japan Earthquake in 2011 and the recent COVID-19 pandemic. Supportive consumption, which entails consumption of products to support producers who are affected by disasters, can be seen as a part of ethical consumption as it considers the sustainability of production areas and producers.

Regarding the relationship between disasters and support consumption, there are studies on the conditions for increasing donations to disaster-stricken areas and consumer altruism and consumer evaluation of foods produced in disaster-affected areas [12,13]. In Japan, studies have also been conducted on the relationship between consumers’ risk aversion and their attitudes toward risk aversion related to the radioactive contamination in Fukushima [14]. However, the relationship between these consumptions and daily consumption trends (e.g., use of co-ops) and presentation of information has not yet been examined. In this study, we examine the relationship between co-op use and disaster-related ethical consumption.

The literature shows that product information significantly influences purchasing behavior and WTP. Uchida et al. [15] (p. 68) showed how information changes consumer perception of a seafood ecolabel, arguing that “valuation for a seafood ecolabel increases only when the information is perceived positively (credible/interesting); whereas exaggerated information (which is perceived as less credible) has insignificant effects on WTP”. Conducting a systematic review of the commercialization of suboptimal food, Hartmann et al. [16] found the importance of framing suboptimal food positively, such as with sustainability and corporate social responsibility messages or by highlighting the products’ “naturalness. [16] (p. 1)” Using eye-tracking technology, Meyerding and Merz [17] demonstrated that visual attention to organic labels on apples is related to consumers’ purchasing behavior. Nonetheless, the influence of daily exposure to information on agricultural production, which is made possible by the co-ops, on consumer valuations remains unclear.

Therefore, this study aims to investigate the extent to which the daily use of co-ops can affect consumers’ WTP regarding the ethical implications of natural disasters on products.

## 2. Materials and Methods

Given the aforementioned considerations, we hypothesized that the presentation of information and the daily use of co-ops each have specific impacts on WTP, as follows:(1)Presenting information about the link between sustainable agricultural production and the purchase of scratched products would enhance consumer evaluation of scratched apples.(2)The daily use of co-ops would increase consumers’ evaluation of scratched apples.

As for the consumer evaluation of unsightly food, studies on suboptimal food have been accumulating in relation to sustainable agriculture [18,19,20]. Huang et al. [21] suggests that providing information on the link between sustainability and food consumption would be effective to encourage consumption of suboptimal foods. In this study, therefore, we present the relationship between sustainable agricultural production and natural disasters as one of the focal points.

Yamoah [8] points out that in addition to price, the tendency of usual consumption is a barrier to ethical consumption. In this study, we consider co-op use as one of the usual consumption tendencies and use the above hypotheses to determine whether the function of co-ops, as pointed out by Kondoh [2] and Wahn [3], positively affects the consumption of suboptimal food.

Apples are taken as a case study because they are often damaged on the surface by natural disasters and are widely consumed fresh in Japan. On average, 9705 g of apples were purchased per household in Japan in 2020–2022 (about 32,300 g apples), making them the second-most consumed fruit in Japan after bananas [22].

We used a choice-based conjoint analysis to estimate consumers’ preferences by asking them to select the products they would like to purchase through questionnaires. Participants were presented with the options (including pictures) of damaged or undamaged apples and were asked to select the one they would prefer to purchase, or neither. An example of the interface presented to the respondents is depicted in Figure 1. Specifically, the respondents were asked to select one fresh, uncut apple of medium size (approximately 300 g). The apples were distinguished based on four characteristics: cultivation method, donation for the recovery of the disaster-affected production area, bruises and scars, and price (tax included). We assumed that the other characteristics (production area, apple variety, etc.) would remain the same across all options.

The first product attribute was “donation to support production area recovery”. As for cultivation method, since the organic cultivation of apples is very difficult and few farmers in Japan practice it, we prepared two options: special cultivation and conventional cultivation. Respondents were given an explanation about these product attributes in the form shown in Table 1 before the options were displayed. In the cross-sectional model, in addition to the commodity attributes noted above, the personal attributes listed in Table 2 were used as explanatory variables. The correlation coefficients of the explanatory variables were low enough to estimate the models.

For product attributes, in addition to the main variables such as price and appearance, we also set cultivation method and donation amount. Previous studies have shown that cultivation methods, including organic cultivation, influence consumers’ WTP [23,24,25]. In the case of apples, special cultivation was adopted as an alternative cultivation method to organic cultivation because fully organic cultivation is technically difficult and organically grown apples are not available in general supermarkets. Studies related to consumption in support of disaster-affected or locally grown products have positioned the amount of donations as an important factor [13,26]. Therefore, we included the amount of donations as an explanatory variable. The price level of apples in the conjoint analysis was set based on the five-year average of the Retail Price Survey conducted by the Ministry of Internal Affairs and Communications in Japan [22]. According to the survey, the national average was 153 JPY, the average of the maximum price was 192.7 JPY, and the average of the minimum price was 111 JPY. The intervals between each level were equalized, and four levels were set at 80, 120, 160, and 200 JPY, taking into account the premiums placed on scratches and other product attributes.

The explanatory variables for individual attributes in Table 2 were set as follows. For the personal attributes, in addition to the variables introduced as personal attributes in general social science estimations, such as income, age, and education, we also included household characteristics such as household demographics (e.g., number of people, elderly households), which, according to previous studies, have a significant influence on the consumption trend [27,28]. Additionally, being engaged in agriculture or the food industry is generally related to the occupation variable included in the personal attributes. Those who are engaged in agriculture or the food industry are more likely than those who are not to routinely obtain information related to food production and distribution. The information asymmetry may be mitigated by the fact that those who are engaged in agriculture and food industry have more opportunities to obtain information related to food production and distribution on a daily basis than those who are not engaged in agriculture and food industry. In the case of Aomori and Nagano, which are the main apple-producing regions in Japan, consumers living in these regions may have more information on apple production than consumers in other regions, and out-of-specification products whose quality does not cover the transportation cost may be sold at local direct sales outlets more frequently than in other regions, thus the supply–demand balance may differ from that in other regions. The possibility that the supply–demand balance differs from that in other regions was taken into account when introducing the variable.

The survey targets were selected from among the monitors owned by Nikkei Research Inc., an Internet research firm, by dividing Japan into six regional blocks and using a stratified sampling method based on the population composition of each region. In this process, we allocated the respondents so that the gender and age groups were as equal as possible. In addition, respondents to the conjoint survey were screened by the following questions. A total of 4100 samples were collected, of which 1702 respondents chose “not the main person to purchase daily groceries”, 53 selected “purchase fresh food less than once a year”, 637 chose “purchase fresh apples less than once a year”, and 11 selected “could not choose which apple to purchase with any confidence at all”. Finally, 1697 responses were deemed suitable for analysis. Table 2 presents descriptive statistics of the personal attributes of the respondents.

To create the product profiles, we used the product attributes in Table 3. Each respondent was shown three cards (two types of apples and a no-buy option) and asked to make a selection eight times. For this, we used the SMRT 4.23.0 software provided by Sawtooth Software. In SMRT, we used “Balanced Overlap”, which is an algorithm for profile creation that accounts for the crossover effects between attributes while improving the efficiency of the calculation. For details, see Sawtooth Software, Inc. [29] (pp. 16–19). The output profile set of 1700 patterns was created by the SMRT and randomly assigned to 1708 respondents, allowing for overlap, and 1061 patterns were ultimately used.

Before answering the choice set selection, half of the respondents saw the following information about relationship between apple production and natural disasters:

“Suppose that several large typhoons hit the apple-growing region just before the harvest, causing scratches on most of the apples produced. In the event of such an unpredictable disaster, consumers can support the producers and production area by buying damaged apples at a certain price, which will help maintain the production area in the long-term”.

Respondents were told that a certain level of damage is inevitable every year and that consumers can support farmers and the sustainability of the production area by purchasing damaged apples for a fair price. This information was intentionally ambiguous yet intuitive, should consumers have difficulty in gathering detailed information at the decision-making stage of the purchase.

The obtained data were analyzed using the mixed logit model developed by Train [30], which has the advantage of relaxing the independent of irrelevant alternatives (IIA) condition and considers the diversity of individual preferences as compared to the conditional logit model analysis by McFadden [31].

The probability *π_in_* that respondent *n* chooses profile *i* is expressed by Equation (1). *V_in_* is the utility gained by selecting profile *i*, expressed by Equation (2). *X_ik_* are the attributes of *k* profiles and *p_i_* is the price of the profile *i*. Profile *j* is the profile not selected by the respondent.

The utility parameter *β* is distributed with probability density *f*(*β*), indicating that *β* is not uniformly distributed across all individuals. This allows us to assume the diversity of preferences.
(1)$$\pi_{in}=\frac{\exp(V_{in})}{\sum_{k=1}^{K}{(V}_{jn})}f\left(\beta \right)d\beta$$
(2)$$V_{in}=\sum_{k=1}^{K}\beta_{ik}x_{ik}+\beta_{ip}p_{i}$$

To ensure robustness of our analysis, we estimated two additional models. The first model was a main effects model which excluded personal attributes as explanatory variables while examining the effect of product attributes. In this model, we assumed normal distributions for all variables. Coefficients of product attributes were set to be random, while coefficients of cross-terms were set to be non-random.

The second model was a cross-term model that included the cross-term of personal attributes x product attributes as explanatory variables. Using the coefficients estimated from this model, we calculated the marginal WTP (MWTP) of the average individual for each attribute, following Hensher et al. [32] (Equation (3)). Although the MWTP by WTP space provides more precise MWTP estimation, we did not use this method in this study to simplify the discussion. See Twain and Week [33] for details.
(3)$${WTP}_{k}={-\beta }_{k}/\beta_{p}$$

## 3. Results

Table 4 shows the estimation results. “Buy (ASC)” is an alternate specific constant (ASC) that represents the value of the apple in the absence of all attributes presented in Table 3, or the value of the apple as evaluated by something other than the attributes. The value is about 211 JPY in Model 1 and 214 JPY in Model 2, which does not deviate greatly from the price of one apple in general. Additionally, we can check the magnitude of the effect of the variable on consumer evaluation by looking at what percentage of this value it is.

First, we discuss the estimation results of the main effects model (Model 1). In the case of pesticide-reduced cultivation and donations to disaster-affected farmers, the valuation increased by 6.7% (about 14 JPY) and 1.8% (about 4 JPY), respectively, compared to 211 JPY for the Buy (ASC). Both variables are significant at the 1% level. The presence of scratches lowered the evaluation by 13.9% (about 29 yen), indicating that the effect of blemishes on consumer evaluations is greater than that of cultivation methods or the presence of donations.

Second, we discuss the results of the cross-term model (Model 2), which considers the differences in individual preferences. The effect of product attributes in the cross-sectional model is similar to that in the main effects model, indicating that there is robustness in the impact of these variables on consumer evaluations. For the interaction term between personal attributes and bruises, being male has a negative effect on the evaluation of bruises. Conversely, having a food-related job and a higher educational level have a positive impact. The presentation of information and daily co-op use, which are of particular interest in this study, are both statistically significant and have a positive impact on the evaluation. Particularly, consumers who use co-ops on a daily basis rated scratched apples 3.6% (about 8 JPY) higher. This suggests that co-ops’ sales methods and public relations activities increase consumers’ understanding of scratched apples. Furthermore, the results of this study show that even simple information, such as that presented here, has the effect of increasing consumers’ evaluation of damaged apples.

## 4. Discussion

This study examined how providing information on disasters and agricultural products affects consumers’ ethical consumption using choice-based conjoint analysis. Our findings revealed that providing simple information and using co-ops daily can change consumer evaluation of damaged apples, particularly for individuals who have relationships with the agricultural/food industry and have a higher educational level.

Consumers generally tended to lower their evaluation of flawed apples as compared to unflawed apples. Although cultivation methods and donations to the affected farmers had the effect of increasing their evaluation of apples, the magnitude of this effect was small compared to the decrease in the evaluation owing to flaws.

However, a cross-sectional model, which allowed the examination of differences in the evaluation of scratched apples by individual attributes, showed that the presentation of simple information and the daily use of co-ops increased consumer evaluation of scratched apples. Although they are relatively fixed attributes, working in the agricultural or food industry or having families work in these industries and having a higher educational level had a positive effect on WTP.

These results suggest that providing simple information can improve consumer evaluation of flawed apples. The information does not have to be precise and complex. Furthermore, co-op users, who are likely to be exposed to producer-related information on a daily basis, have a relatively higher understanding of agricultural production and natural disasters. These results support our initial hypothesis.

The results of the present analysis of apples can be applied to perishable products that are damaged or deformed by disasters, or other appearance-only defects that do not affect the taste or other aspects of the product. The results of this study bridge previous research on disasters and supportive consumption with research on the effects of daily consumption propensities on product evaluations. We also quantified the effects of Alternative Food Network (AFN)-based co-op studies on co-op use and its contribution to sustainable agricultural production, which was ambiguous in the AFN-based co-op studies [2,3]).

Regarding specific recommended measures, this study indicates that displaying simple explanatory notes in stores at points of purchase or through other means following a disaster and disseminating producer information daily may contribute to sustainable agricultural production.

This study has several limitations. First, we equated using a co-op with being informed about agricultural production. Second, in our survey, co-op members and non-co-op members who only buy from co-op stores were treated as identical if they both used co-ops on a daily basis. To better understand the role of co-ops, it is necessary to analyze in further detail how these individuals are involved in the co-ops, such as whether they usually participate in study groups on agricultural production or are involved in product development. In addition, it is necessary to analyze not only the appearance of the products affected by disasters, but also more diverse cases, such as those in which the quality of the products, such as their perishability, is damaged, or in the case of processed raw materials.

## Figures and Tables

**Figure 1 foods-12-02498-f001:**
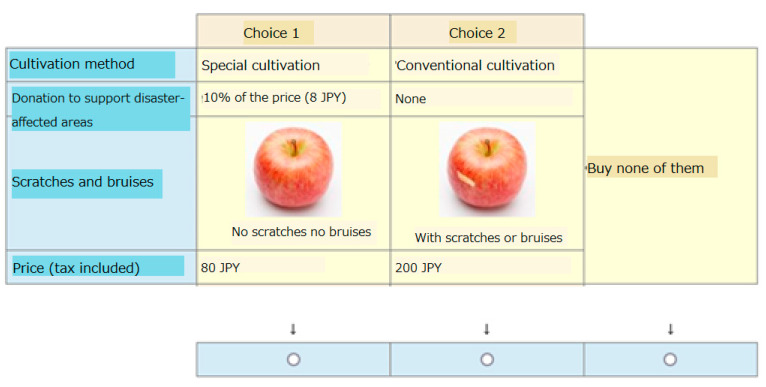
An example of the choice set shown to the respondents.

**Table 1 foods-12-02498-t001:** Descriptions of the product attributes shown to the respondents.

Attributes *	Description
Cultivation method	There are two types of cultivation methods: “conventional cultivation”, in which apples are grown using pesticides and chemical fertilizers at a level where safety is fully confirmed, and “special cultivation”, in which apples are produced by reducing the use of chemical fertilizers and pesticides by 50% or more compared to conventional cultivation. Specially cultivated agricultural products are produced under guidelines set by the government and have passed the government’s certification system.
Donation to support the disaster-affected area	This indicates that a portion (5% or 10%) of the price you pay for the apple will be used to support food producers (mainly farmers) affected by natural disasters, such as typhoons and earthquakes. This donation is not an additional payment you make on your food purchase but is included in the product’s price (tax included).
Scratches and bruises	This indicates that there are scratches and/or bruises resulting from typhoons or other natural disasters. Although they may not look good, the taste and safety of the product are the same as that of a product with no blemishes. However, early consumption is recommended as spoilage may occur from the bruises.
Price (tax included)	This indicates the amount you will pay for the food and its consumption tax. Shipping and handling charges are not included.

* Characteristics other than those mentioned above are assumed identical for all choices.

**Table 2 foods-12-02498-t002:** Descriptive statistics of personal attributes.

Variables	Description	% of 1
male	male = 1, female = 0	46.67
aomori	living in Aomori = 1, others = 0	0.53
nagano	living in Nagano = 1, others = 0	1.59
foodjob	the respondent or his/her family members are engaged in agriculture or food industry = 1, others = 0	9.02
livalone	living alone = 1, others = 0	21.26
elder	living with elderlies older than 75 = 1, others = 0	22.77
hiincome	household income > 10 million JPY = 1, others = 0	19.32
college	graduate from a 4-year college or graduate school = 1, others = 0	57.67
info	with the information about apple production and natural disasters= 1, no information = 0	51.27
coop	use co-op almost every week = 1, others = 0	16.21

**Table 3 foods-12-02498-t003:** Product attributes and attribute levels.

Cultivation Method	Donation to Support Disaster-Affected Area	Scratch	Price (Tax Included)
Special cultivation (reduced use of pesticides and chemical fertilizers)	None	Yes	80 JPY
	5% of the price	No	120 JPY
	10% of the price		160 JPY
Conventional cultivation			200 JPY

**Table 4 foods-12-02498-t004:** Estimation results.

Variables	Model1	MWTP1	Model2	MWTP2
Buy (ASC)	8.03 ***	210.8	8.39 ***	213.58
price	−0.04 ***		−0.04 ***	
cultivation method	0.54 ***	14.12	0.51 ***	12.85
donation	0.15 ***	3.85	0.14 ***	3.44
scratch	−1.11 ***	−29.20	−1.28 ***	−32.45
male × scratch			−0.27 **	−6.82
aomori × scratch			0.10	
nagano × scratch			0.17	
foodjob × scratch			0.46 ***	11.7
livalone × scratch			−0.15	
elder × scratch			0.07	
hiincome × scratch			0.01	
college × scratch			0.21 *	5.23
info × scratch			0.25 **	6.3
info × donation			0.06	
coop × scratch			0.30 **	7.74
No. of obs.	40,728 (1697)		35,352 (1473)	
AIC	17,064.51		14,530.84	

*** Statistically significant at the 1% significance level; ** Statistically significant at the 5% significance level; * Statistically significant at the 10% significance level.

## Data Availability

All related data and methods are presented in this paper. Additional inquiries should be addressed to the corresponding author.

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
