# Peer review of "Consumer Evaluation of Agricultural Products Produced in Areas Affected by Natural Disasters: A Case Study of Damaged Apples in Japan"

_foods, 2023, doi:10.3390/foods12132498_

Round 1

Reviewer 1 Report

This article reports a study on how information given to consumers may affect their willingness to pay for damaged agricultural food products, using the case of apples.

The topic is of interests because damages regularly occur without affecting the taste or the food quality, except for its visual appearance, but result in declassification of the product and lower price, hence lower income for the farmers. Understanding how it is possible to increase consumers' acceptance of such products is therefore of interest.

The methods are clearly presented (see attached manuscript for some comments), so are the results.

However, the weak part of the manuscript is the discussion. Indeed, most of the discussion is a (second) presentation of the results, and do not challenge these results in view of 1) authors' hypotheses and 2) other works. It is therefore difficult to draw conclusions from this presentation. How general are the results? Could they be applied to other agricultural products? etc.

Strengthening the discussion section is therefore necessary.

Author Response

Thank you for your review and recommendations.

Please see the attachment for my revision.

Reviewer 2 Report

The article is interesting and presents important issues of apple consumer choices in Japan. Methodologically, the article is prepared correctly. The appropriate methodology allows conclusions to be drawn. The main comments are:

1. The article deals with the specifics of the Japanese market and some issues are incomprehensible to readers from other countries - especially European ones. The issue of 'consumer cooperatives' (co-ops) needs wider clarification. 1) Who is a member of such a cooperative? How does the cooperative operate? What is the purpose of such cooperatives? How much are the membership fees? Is it a common legal form in Japan?

2. An important element determining the willingness to purchase a certain product at a higher price, besides awareness, is the amount of income generated by consumers. I believe that this issue would need to be presented in more detail. Why were only two levels of consumer income chosen? What is the expenditure of citizens in Japan on food in the household budget? What is the per capita consumption of apples in Japan?

3. I think the 'discussion' point would need to be presented more widely. There is also a lack of clear conclusions from the research.

Author Response

(The authors gave the same response as above.)

Reviewer 3 Report

Please see the attached document for details of my comments.

The English is fine but you might want to include explanation  for specific term like the co-op system.

Author Response

(The authors gave the same response as above.)

Round 2

Reviewer 3 Report

Thank you for replying to my comments. I think now most of the concerns have been answered and modified. After removing or including the footnote added in the paper it is acceptable.

After some minor editing, the paper is ready to be published.

Author Response

Thank you very much for your comments.
I have moved the footnote1 on page7, which is regarding the estimation of the MWTP, to the main text.
I hope this correction will fit the publication.